# Enhancing Sumoylation Site Prediction: A Deep Neural Network with Discriminative Features

**DOI:** 10.3390/life13112153

**Published:** 2023-11-02

**Authors:** Salman Khan, Mukhtaj Khan, Nadeem Iqbal, Naqqash Dilshad, Maram Fahaad Almufareh, Najah Alsubaie

**Affiliations:** 1Department of Computer Science, Abdul Wali Khan University, Mardan 23200, Pakistan; salman@awkum.edu.pk (S.K.); nikhan@awkum.edu.pk (N.I.); 2Department of Information Technology, The University of Haripur, Haripur 22620, Pakistan; mukhtaj.khan@uoh.edu.pk; 3Department of Convergence Engineering for Intelligent Drone, Sejong University, Seoul 05006, Republic of Korea; dilshad.naqqash@gmail.com; 4Department of Information Systems, College of Computer and Information Sciences, Jouf University, Sakaka 72388, Saudi Arabia; mfalmufareh@ju.edu.sa; 5Department of Computer Sciences, College of Computer and Information Sciences, Princess Nourah bint Abdulrahman University (PNU), P.O. Box 84428, Riyadh 11671, Saudi Arabia

**Keywords:** deep neural network, sumoylation sites, machine-learning algorithm, half-sphere exposure, artificial intelligence

## Abstract

Sumoylation is a post-translation modification (PTM) mechanism that involves many critical biological processes, such as gene expression, localizing and stabilizing proteins, and replicating the genome. Moreover, sumoylation sites are associated with different diseases, including Parkinson’s and Alzheimer’s. Due to its vital role in the biological process, identifying sumoylation sites in proteins is significant for monitoring protein functions and discovering multiple diseases. Therefore, in the literature, several computational models utilizing conventional ML methods have been introduced to classify sumoylation sites. However, these models cannot accurately classify the sumoylation sites due to intrinsic limitations associated with the conventional learning methods. This paper proposes a robust computational model (called Deep-Sumo) for predicting sumoylation sites based on a deep-learning algorithm with efficient feature representation methods. The proposed model employs a half-sphere exposure method to represent protein sequences in a feature vector. Principal Component Analysis is applied to extract discriminative features by eliminating noisy and redundant features. The discriminant features are given to a multilayer Deep Neural Network (DNN) model to predict sumoylation sites accurately. The performance of the proposed model is extensively evaluated using a 10-fold cross-validation test by considering various statistical-based performance measurement metrics. Initially, the proposed DNN is compared with the traditional learning algorithm, and subsequently, the performance of the Deep-Sumo is compared with the existing models. The validation results show that the proposed model reports an average accuracy of 96.47%, with improvement compared with the existing models. It is anticipated that the proposed model can be used as an effective tool for drug discovery and the diagnosis of multiple diseases.

## 1. Introduction

Sumoylation is a post-translation modification (PTM) mechanism involved in many critical biological processes, including gene expression, protein localization and stabilization, and genome replication. Sumoylation is an enzyme-mediated covalent modification of a protein sequence in which a chemical group is added to a specific location [1]. It modifies the activity of most eukaryote proteins. These modifications include nitrosylation, phosphorylation, lipidation, proteolysis, ubiquitination, glycosylation, methylation, and acetylation [2,3]. These modifications significantly increase the variety of possible protein final forms generated using the same genetic sequence [4]. Moreover, the analysis of these modifications results in mass changes in proteins; however, their identification generates vital insight into biological function.

Recently, 450 new distinctive protein variations have been recognized, i.e., ubiquitination [5], acetylation [6], phosphorylation [7], and sumoylation [8]. These distinctive protein modifications can modify protein interactions, intracellular distribution, activity of target proteins, and protein longevity through PTM [9]. Among all these protein modifications, the sumoylation sites have received great attention from the research community because the sumoylation sites are widely used for post-translational protein modification. The sumoylation sites play an important role in the functional proteomic [10]. For example, it expands genetic code, underlies protein recycling and degradation, localizes proteins within the cell, regulates the activity of cellular physiology, activates or deactivates transporters and enzymes, and interacts with other cellular molecules. In addition, recent studies have shown that numerous diseases and disorders, i.e., Parkinson’s diseases and Alzheimer’s, have been found to be closely related to the sumoylation site. To extract the structure information, a series of bioinformatics tools have been developed [11,12,13]. Given the importance of sequential and structural bioinformatics in modeling drug discovery, computational biology has played a significant role [14]. Therefore, The discovery of sumoylation sites in proteins is crucial not only for gaining a thorough understanding of many critical biological processes but also for the creation of effective medications that can be utilized as a possible therapeutic target for cancer and other disorders [15].

Given the significance of sumoylation sites in the genome, detecting and identifying them has become a crucial research area in computational biology and bioinformatics. Several models have been proposed in the literature for the identification of sumoylation sites. These models include SUMOsp [16], SUMOsp2.0 [17], and GPS-SUMO [18]. However, the mentioned models are developed based on the group-based phosphorylation scoring algorithm (GPS). The GPS is a simple and intuitive approach that calculates peptide similarities and decides which peptides are closest to the given peptides after clustering known peptide groups [19]. Similarly, Xu et al. [20] presented another technique, SUMO_LDA, developed by merging three distinct types of sequence characteristics into the general pseudo-amino acid composition (PseAAC). The proposed model generated quite promising results; however, the SUMO_LDA is based on linear discriminant analysis (LDA), which suffers from several problems. First, the LDA method fails to find the lower-dimensional space if the number of features is higher than the number of samples in the feature vector, which leads to the problem of small sample size (SSS) [21]. Second, the LDA algorithm faces the linearity problem if the classes are non-linearly separable. In this case, the LDA algorithm will be unable to discriminate the classes accurately [22].

Recently, Xu et al. [23] and Chen et al. [24] proposed computational models known as SUMOPre and SUMOhydro, respectively. The proposed predictors used SVM as a learning algorithm for the classification of SUMO sites. Similarly, Jia et al. [25] developed the pSumo-CD predictor. The proposed predictor employed the general PseAAC method and SVM and RF algorithms for the classification and identification of sumoylation sites. Most recently, Sharma et al. [26] proposed HseSUMO for the prediction of sumoylation sites. The HseSUMO employed the half-sphere exposure method and decision tree algorithm as the learning method. The above-mentioned models have produced promising results; however, conventional learning models require significant human expertise to extract prominent features automatically [27]. Additionally, the prediction performance of these models is limited in the case of nonlinearity in the dataset.

Hence, a more influential and vigorous computational predictor is required in the field of bioinformatics for the identification of sumoylation sites. This paper suggests a predictor for the accurate identification of sumoylation sites using an optimized deep-learning algorithm with efficient feature-extraction methods known as Deep-Sumo. First, the proposed predictor applies half-sphere exposure (HSE) measures as a feature-extraction method. Second, to select efficient features by eliminating irrelevant and noisy features, an unsupervised PCA method is applied. Finally, a multilayer DNN method is functional as a classifier. The proposed model performance is rigorously examined based on diverse performance measurement metrics using a 10-fold cross-validation method. The experiments showed that the proposed predictor attained an average prediction accuracy of 96.47%. Furthermore, a comparison with recently published predictors revealed that the proposed Deep-Sumo outperformed the existing predictors regarding accuracy and other performance measurement metrics. The major contributions of this paper are as follows:An intelligent and robust computational model built on multiple layers.A deep model with automatic weight optimization using standard learning procedure for the accurate prediction of sumoylation sites.The employment of the half-sphere exposure method that efficiently transforms peptide sequences into a feature vector.Efficient features extraction by removing noisy and irrelevant features using an unsupervised PCA algorithm.Mathew’s Correlation Coefficient (MCC), accuracy, sensitivity, specificity, and area under the ROC Curve (AUC) are used to evaluate the performance of the proposed model.

## 2. Proposed Model Design

This section presents the prototype and design of the proposed model. Its architecture is depicted in Figure 1. The different components of the model are explained in detail below:

### 2.1. Benchmark Dataset

A legitimate and trustworthy dataset is always required to generate an effective and dependable computational model [28,29]. Therefore, we decided to use a reliable benchmark dataset from [26] for training and verification of the results of the suggested model. The chosen benchmark dataset may be visualized as shown below:(1)$$S=S^{p}\;U\;S^{N}$$
where S  represents sumoylation and non-sumoylation site sequences. $S^{p}$ and $S^{N}$ represent a subgroup of sumoylation samples and non-sumoylation site samples, respectively. The dataset used in this study was taken from the database Compendium of Protein Lysine Modification (CPLM) [30], a resource that manually collects details of lysine PTMs of 12 different forms. The original benchmark dataset contained 780 sumoylation site sequences (i.e., positive samples) and 21,353 non-sumoylation site sequences (i.e., negative samples). The benchmark dataset needs to be more balanced. A classification problem occurs when the categorization is biased towards the majority class in an unbalanced dataset. The literature has studied several data and algorithm-level strategies for addressing the class imbalance problem and the problem of the unbalanced dataset. However, a data-level approach, which includes oversampling and undersampling approaches, is extensively used [31]. The undersampling strategy can often provide a modest answer, but the oversampling method may raise the danger of model overfitting owing to data sample repetition. Therefore, to balance the benchmark dataset in this work, we used the undersampling approach with the NearMiss algorithm [32] developed in the Python programming language. After employing the undersampling method, we obtained a balanced benchmark dataset, which contained 780 negative samples and 780 positive samples. Additionally, we divided the data randomly into 10 groups: 9/10 (i.e., 702 positives and 702 negatives) as the cross-validation dataset and 1/10 (i.e., 78 positives and 78 negatives) as the independent test dataset. The data frame sample function in the pandas Python library was used for random number generation [33].

### 2.2. Feature Formulation Technique

The most fundamental and difficult step in computational biology is a discrete model or vector representation of a given sequence [34]. The discrete model should preserve the structure information and keep the key potential characteristics of the given sequence.

This paper used HSE as a feature-extraction method—a two-dimensional solvent exposure measure developed by Hamelryck [35]. Further solvent exposure measures, i.e., RD [36], rASA [37], ASA [38], and CN [39], have drawbacks compared to the HSE method. Moreover, the HSE method reported superior performances than the existing exposure measures in terms of conservation among different folds, computational complexity, protein stability, sensitivity, predictability, and computational speed.

The HSE method divides the sphere of a residue into two half-spheres, i.e., the first half-sphere is HSEβ or HSE-up, and the second half-sphere is referred to as HSEα or HSE-down, as shown in Figure 2. The HSE-up and HSE-down represent the upper and lower spheres concerning the residue’s chain side. These two half-spheres possess distinct geometries and energy properties, exhibiting interesting characteristics for characterizing the residue’s spatial neighborhood. We employed a sphere radius of rd = 13 Å for all residues to calculate the measurements. This resulted in a feature vector of 124 features, representing the protein sequence in a discrete model.

### 2.3. Feature Selection

The feature vector’s presence of noisy, duplicated, or irrelevant information may severely impact the classifier performance. Therefore, we use Principal Component Analysis (PCA) for feature selection. The PCA algorithm is a multivariate data-processing technique that computes eigenvectors and covariance matrices to decrease the dimensionality of feature vectors.

Consider an input feature vector S, i.e., 124 × 1560, and “k” characterizes the number of desired features.
(2)$$\left[s_{1},s_{2},s_{3},\ldots s_{n}\right]$$

Using the PCA algorithm,

Step 1: Feature vector mean value using Equation (3)
(3)$$\overset{\_}{S}=\frac{1}{i}\sum_{n=1}^{i}s_{n}$$

Step 2: Subtract the mean value using Equation (4)
(4)$$S_{n}^{\prime }=s_{n}-\overset{\_}{s}$$

Step 3: Covariance matrix:(5)$$C_{n}=S_{n}^{\prime }.(S_{n}^{\prime }{)}^{T}BB^{T}$$
where $B=\left\{S_{1}^{\prime },S_{2}^{\prime },\ldots ,S_{p}^{\prime }\right\}(i*j)$ and transposition of a matrix, *T*.

Step 4: The Eigenvalues of a matrix are computed in the fourth step. The eigenvalue values γ, of the covariance matrix can be solved using Equation (6).
(6)$$\det((\gamma I-C_{n})=0$$
where det represents the determinant of the matrix, and I represents the identity matrix. The first eigenvalue must be higher than the second eigenvalue.
$$\left\{\gamma_{1}>\gamma_{2}>\gamma_{3}\ldots \gamma_{n}\right\}$$

Step 5: Eigenvector:(7)$$c_{n}:\left\{\sigma_{1},\sigma_{2},\sigma_{3}\ldots \sigma_{n}\right\}$$

Last step: we choose k values with max eigenvalues and select the eigenvector with the maximum eigenvalue. To attain a high level of classification accuracy in our study, our empirical findings demonstrate that utilizing 80 components effectively captured the essential information within the dataset for feature representation. Interestingly, we observe that increasing the number of features beyond 80 has a detrimental effect on the overall performance of the classification model. Moreover, we carefully chose a subset of 80 features that not only sufficed for accurate classification but also helped prevent overfitting and noise, ultimately leading to better model performance.

### 2.4. Deep Architecture

DNN is an ML algorithm inspired by the human brain [40,41] that includes input, output, and multiple hidden layers in its network topology, as shown in Figure 3.

The DNN’s hidden layers are crucial for learning, but increasing their number can lead to computational costs, overfitting, and model complexity [42,43]. DNNs automatically extract relevant features from unstructured or unlabeled data, making them superior to traditional learning methods. DNNs have been applied successfully in numerous fields, i.e., natural language processing, bio-engineering, and image and speech recognition [44].

As shown in Figure 3, this work predicts sumoylation sites using a benchmark dataset and a DNN model with four hidden layers. The DNN design employs several neurons to assess the input feature vector and create output using Equation (8). The weight matrix for each neuron is established using the Xavier function [45], which may keep the variance constant through each layer. In addition, the weight matrix is updated using a back-propagation approach to eliminate errors between the output and target classes. The input layer and buried layers both use the Tanh nonlinear activation function. A dataset’s nonlinearity and complicated patterns can be learned by a model with the aid of the activation function. Additionally, it decides whether a neuron may be fired or ignored based on the output that a specific neuron produces [46]. A SoftMax activation function is also used at the output layer to provide a value between [0, 1] that reflects the probability that a certain class of data points would be included in the output.
(8)$$y_{a}=f(B_{a}+\sum_{b=1}^{m}x_{b}w_{b}^{a})$$
(9)$$f(i)=\frac{e^{i}}{1+e^{i}}$$

The DNN has confines, including high computational complexity/cost and a need for many computational resources to configure a complex model. Additionally, many training samples are required for effective model training.

## 3. Performance Evaluation

Statistical machine-learning models must be evaluated using performance metrics before deployment in a real production environment. Accuracy is an important metric, but it alone is insufficient. Various performance metrics have been proposed, and the choice of metrics depends on the specific application and problem domain [47,48]. Performance measures include sensitivity (SN), specificity (SP), the area under the curve (AUC), characteristic (ROC), and MCC. As previously stated, this study uses performance measurements to evaluate how well the proposed Deep-Sumo performs because these metrics have been widely utilized in various publications. The performance measurement metrics can be calculated as follows:(10)$$ACC=1-\frac{S_{-}^{+}+S_{+}^{-}}{S^{+}+S^{-}}$$
(11)$$SP=1-\frac{S_{-}^{+}}{S^{+}}$$
(12)$$SN=1-\frac{S_{+}^{-}}{S^{-}}$$
(13)$$MCC=\frac{1-(\frac{S_{-}^{+}\;+\;S_{+}^{-}}{S^{+}\;+\;S^{-}})}{\sqrt{(1+\frac{S_{-}^{+}\;+\;S_{+}^{-}}{S^{+}}})(1+\frac{S_{-}^{+}\;+\;S_{+}^{-}}{S^{-}})}$$

## 4. Results and Analysis

### 4.1. Hyper-Parameter Analysis

The motivation of this section is to obtain optimum configuration values for the hyperparameters used in DNN topology. The important hyperparameters in a DNN include the number of layers and neurons, seed, regularization techniques l1 and l2, activation function, weight initialization, momentum, dropout, updater, iteration, learning rate, and optimizer, as shown in Table 1. These hyperparameters play a crucial role in determining the performance and behavior of the neural network. For example, the number of layers and neurons per layer refers to the architecture of a neural network and can affect the network’s capacity to learn as well as its ability to fit the training data. A seed is a predetermined starting point used to initialize or control random processes, ensuring reproducibility in experiments or computations. Regularization techniques like L1 and L2 regularization help prevent overfitting by adding penalty terms to the loss function. Activation functions are used in each neuron to introduce nonlinearity into the model. The initialization of weights refers to the process of setting the initial values for the parameters (weights and biases) of the network’s neurons before training, and momentum enhances the optimization process by incorporating past gradient information to accelerate convergence and improve stability during training. Dropout is a regularization technique where a random fraction of neurons is dropped during each training iteration. Finally, the “updater” is the mechanism responsible for adjusting model parameters, “iteration” represents a single cycle through the training data, “learning rate” controls the size of weight updates, and the “optimizer” is the overarching algorithm that guides the optimization process by determining how weights are updated in each iteration. To analyze the performance of the DNN on different hyperparameters, we used a grid-search technique that applies different combinations of parameters. For performance analysis, we considered only those hyperparameters that largely affect the performance of the DNN model. These parameters include activation function, learning rate, and number of iterations.

We conducted experiments to examine the effects of activation functions and LR. The results of the experiments are given in Table 2. From the table, it can be observed that the highest accuracy, i.e., 96.47%, is obtained by the DNN classifier at a learning rate value of 0.1 using Tanh as an activation function. Furthermore, the DNN model is continuously improved by decreasing the learning rate; however, after reducing the LR (i.e., 0.09 and 0.08), the DNN model accuracy could not be significantly improved. Hence, the DNN model presented a high accuracy at a learning rate of 0.1 with the Tanh activation function.

Second, we performed several experiments to analyze the DNN performance from the aspect of various iteration numbers at the model training stage. From the experimental results, at 600 iterations, the error losses have become stable using the Tanh, ReLU, and Sigmoid activation functions.

### 4.2. Cross-Validation Scheme Performance Analysis

Statistical learning models in computational biology and bioinformatics can be assessed by validation tests, i.e., jackknife, k-fold, and sub-sampling tests. Among these, k-fold cross-validation is broadly implemented and provides unbiased outcomes. This study analyzed the proposed method’s performance using 10-fold and 5-fold CV tests. The results in Table 3 indicate that the Deep-Sumo model achieved higher accuracy (91.99%) with a 10-fold CV compared to a 5-fold CV (91.03%).

The feature vector obtained through the HSE method contained 124 features that may comprehend inappropriate, redundant, and deafening features. To obtain efficient features and reduce the dimensionality, we have employed the feature-selection method, as discussed in Section 2.3. We reduced the feature vector dimension from 124 × 1560 to 80 × 1560. The evaluation of the proposed model includes the assessment of its performance using both comprehensive and optimized feature sets, ensuring a thorough analysis of its capabilities. The experimental results of this evaluation are shown in Table 3, which shows that the proposed model performance has better quality using an efficient feature set in comparison to the entire feature set. It can be seen from Table 3 that the accuracy of the proposed model is pointedly enhanced in terms of yielded accuracy—91.99% in the case of the entire feature set—and it obtained an average accuracy of 96.47% in the case of the efficient feature set. Similar improvement is reported in the case of other performance measurement metrics, i.e., specificity (96.25%), sensitivity (96.71%), and MCC (0.929), utilizing efficient feature sets in comparison to the entire feature set.

The Deep-Sumo model performance was evaluated using the AUC metric, which measures the accuracy of binary classifiers. Higher AUC values indicate better performance. Figure 4 shows that the Deep-Sumo model achieved the highest AUC values of 0.984 with 10-fold cross-validation and 0.972 with 5-fold cross-validation when using the efficient feature set. These results confirm the superior prediction of the proposed model performance with a 10-fold CV and selected features.

### 4.3. Performance Comparison of Different Classifiers

In this section, a comprehensive comparison is conducted between the performance of the Deep Neural Network (DNN) model and other commonly employed machine-learning algorithms, specifically, K-Nearest Neighbor (KNN), Random Forest (RF), and Support Vector Machine (SVM). The KNN algorithm, a non-parametric instance-based learning method, is utilized in image processing to classify instances based on their Euclidean distances [40,41,43,48,49]. Contrarily, RF is a supervised learning method frequently used for regression and classification problems. The bootstrap approach produces numerous decision trees based on random sample selections. This analysis sheds light on each approach’s advantages and disadvantages. The RF algorithm is a supervised learning algorithm widely used for classification and regression problems, which employs a bootstrap algorithm on training samples to create several decision trees based on random selection. The SVM algorithm is a powerful classification algorithm used in bioinformatics that computes an optimal hyperplane to distinguish among groups for linear and nonlinear classification. To achieve a fair comparison across all learning algorithms, we used the same efficient feature set, performance measurement standards, and validation technique.

Table 4 offers a performance comparison of several algorithms. The table shows that the DNN model performed better than the competing models. For instance, the DNN model achieved an average accuracy of 96.47%, while the RF model achieved the next highest accuracy of 94.10%. Similarly, in terms of MCC, which reflects the stability of a model, the DNN model achieved the highest value of 0.929, surpassing the next highest value of 0.883 obtained by the RF algorithm. The KNN model exhibited the worst performance in the case of all performance measurement metrics.

The analysis of the results suggests two foremost causes for the superior performance of the proposed model compared to conventional learning algorithms. First, traditional learning methods typically employ a single-stack processing layer, which may not effectively handle complex datasets with high nonlinearity. In contrast, to process complex datasets effectively, the DNN model utilizes multi-stack processing layers and automatically extracts optimal features. Furthermore, the performance of the DNN model was compared to conventional learning algorithms in terms of AUC (area under the ROC curve), as shown in Figure 5. The figure illustrates that the proposed DNN model achieved the highest AUC value compared to the conventional learning algorithms. For instance, the DNN model achieved an AUC value of 0.984, while the RF, SVM, and KNN algorithms achieved AUC values of 0.963, 0.920, and 0.902, respectively.

### 4.4. Existing Model Performance Comparisons

Deep-Sumo’s performance was compared to current predictors described by the authors in [26], and the comparative results are shown in Table 5. Deep-Sumo clearly outperformed the previously released predictors, according to the data. For example, the suggested DNN model obtained the best accuracy of 96.47%, whereas the current predictor (HseSUMO) had the second-highest success rate of 89.50%. Similarly, Deep-Sumo had the greatest MCC value of 0.929, whereas HseSUMO obtained an MCC score of 0.790. These findings demonstrate that Deep-Sumo outperformed the current models, with an average success rate improvement of 7.59%.

### 4.5. Performance Comparison on Independent Dataset

To ensure the stability and reliability of the proposed Deep-Sumo, we perform an independent dataset test. The outcome of the Deep-Sumo on an independent dataset in comparison with other learning models is given in Table 6. As shown in Table 6, Deep-Sumo performed remarkably well and reported the highest accuracies, i.e., 92.23%. Similarly, Deep-Sumo reported better sensitivity, specificity, and MCC in comparison with other learning models, as shown in Table 6. Furthermore, in the case of conventional classifiers, RF reported the highest accuracies, i.e., 91.43%, in comparison with SVM and KNN.

## 5. Conclusions

This paper proposed Deep-Sumo, an intelligent computational model for the prediction of sumoylation sites. The proposed model applied the half-sphere exposure method to depict protein samples into a feature vector, and it employed the DNN model as a prediction engine. Additionally, an unsupervised PCA algorithm was employed for optimum feature selection, and a data-level (i.e., undersampling) method was employed to balance the benchmark dataset. The performance of the proposed model was extensively evaluated, and experimental results showed that the Deep-Sumo model accurately predicts the sumoylation sites. Moreover, the performance of the proposed Deep-Sumo was compared with the existing model, and the comparison analysis yielded that Deep-Sumo outperformed the existing models.

It is proposed that the suggested method can be an accessible prediction method for the identification of sumoylation sites. Moreover, it could be utilized for drug discovery and basic research. In future work, as mentioned in a series of recent publications [50,51], we plan to develop a publicly accessible web server for Deep-Sumo, allowing experimental scientists to use it to identify sumoylation sites easily. In addition, a huge amount of genome data is generated due to advancements in next-generation sequencing technology, which poses computational challenges for sequential computing approaches. We also plan to apply parallel programming techniques to parallelize computations on the number of processing nodes using big-data analytics platforms [52]. For the convenience of researchers, the source code of the proposed predictor, along with the dataset, is available at https://github.com/salman-khan-mrd/Deep-Sumo (accessed on 8 September 2023).

## Figures and Tables

**Figure 1 life-13-02153-f001:**
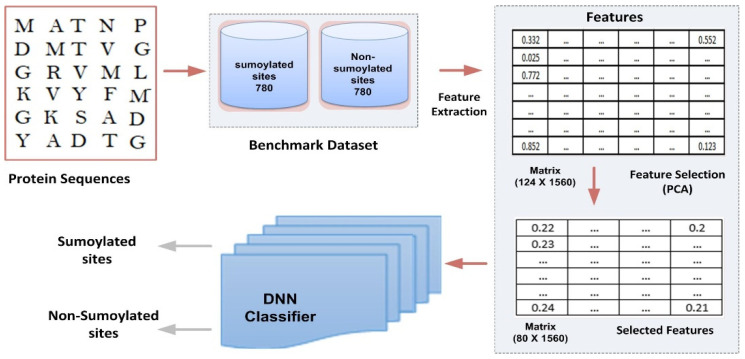
Proposed computational model framework.

**Figure 2 life-13-02153-f002:**
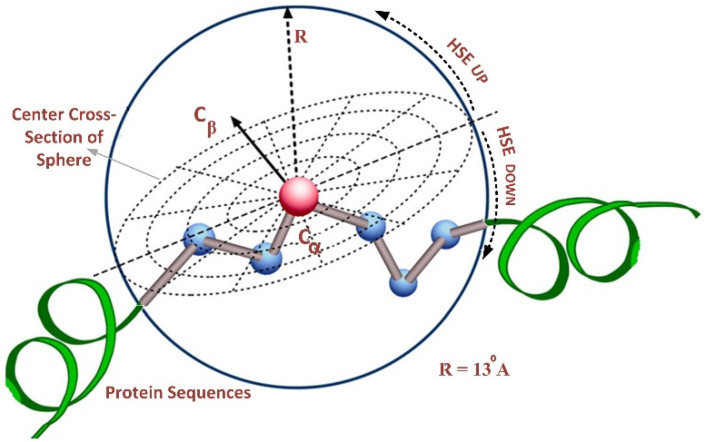
Overall structure of half-sphere exposure [26].

**Figure 3 life-13-02153-f003:**
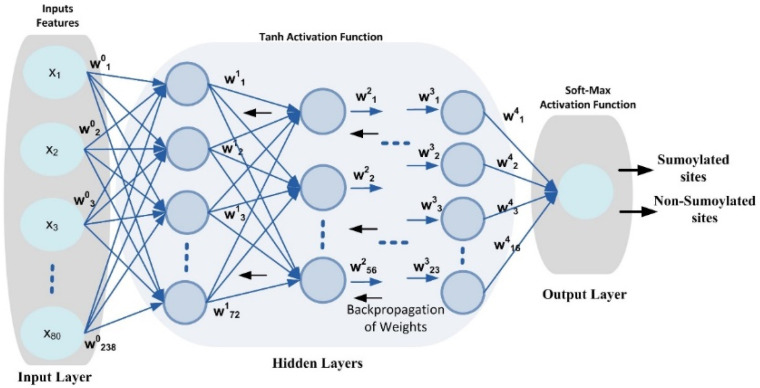
DNN configuration topology.

**Figure 4 life-13-02153-f004:**
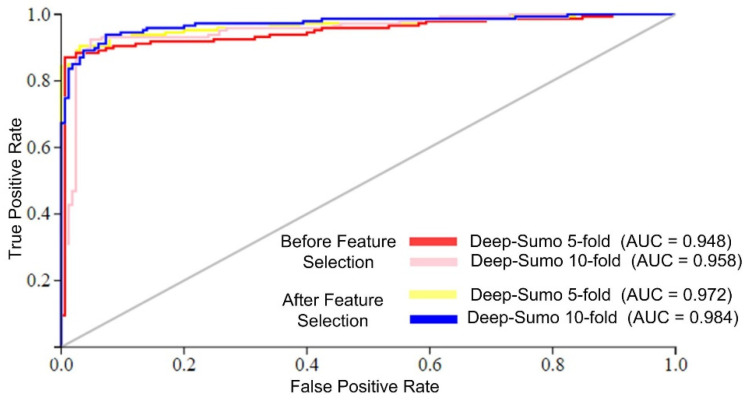
AUC of different cross-validation schemes.

**Figure 5 life-13-02153-f005:**
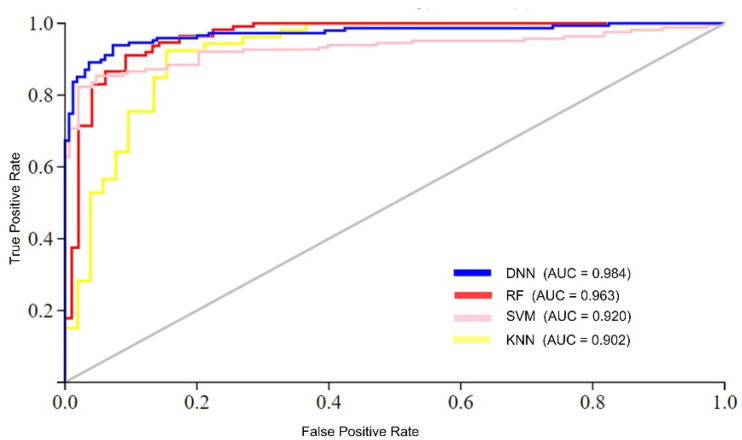
AUC performance comparison of ML algorithms.

**Table 1 life-13-02153-t001:** DNN model optimum hyperparameter values.

List of Parameters	Optimal Values
Neurons at hidden layers	80-72-56-23-16-2
Seed	12345L
Regularization l2	0.001
Activation Functions	Tanh and SoftMax
Weight initialization function	XAVIER function
Momentum	0.9
Dropout	0.25
Number of hidden layers	4
Updater	ADAGRAD function
Training iterations	600
Optimizer	SGD Method
LR	0.1

**Table 2 life-13-02153-t002:** Grid search of the DNN model.

LR	Tanh	Sigmoid	ReLU
0.08	96.47	89.23	92.01
0.09	96.47	89.23	92.01
0.1	96.47	89.23	92.01
0.2	93.41	87.78	90.71
0.3	91.72	85.80	89.21
0.4	90.35	84.17	87.84
0.5	87.97	82.46	86.44
0.6	85.97	80.74	85.04
0.7	83.96	79.03	83.64
0.8	81.95	77.31	82.24
0.9	79.95	75.60	80.84

**Table 3 life-13-02153-t003:** Performance evaluation of the Deep-Sumo model using the entire feature set and efficient features set.

Method	ACC (%)	SP (%)	SN (%)	MCC
Deep-Sumo (5-fold)	91.03	85.21	95.88	0.821
Deep-Sumo (10-fold)	91.99	89.44	94.70	0.841
**After Feature Selection**				
Deep-Sumo (5-fold)	95.19	93.04	97.40	0.905
Deep-Sumo (10-fold)	96.47	96.25	96.71	0.929

**Table 4 life-13-02153-t004:** Proposed model comparison with other ML.

ML	SP (%)	MCC	ACC (%)	SN (%)
Deep-Sumo	96.25	0.929	96.47	96.71
SVM	93.08	0.802	90.00	86.92
RF	95.90	0.883	94.10	92.31
KNN	90.01	0.801	89.10	78.33

**Table 5 life-13-02153-t005:** Existing model performance comparisons.

Method	ACC (%)	SP (%)	SN (%)	MCC
pSumo-CD	72.80	92.10	53.60	0.494
HseSUMO	89.50	89.50	89.50	0.790
Deep-Sumo	96.47	96.25	96.71	0.929

**Table 6 life-13-02153-t006:** Proposed model comparison with other ML on an independent dataset.

ML	ACC (%)	SP (%)	SN (%)	MCC
Deep-Sumo	92.23	93.53	90.93	0.892
RF	91.43	92.12	90.74	0.854
SVM	88.32	91.76	84.88	0.782
KNN	87.65	88.12	87.18	0.781

## Data Availability

Dataset will be provided on request.

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
