# Peer review of "Enhancing Sumoylation Site Prediction: A Deep Neural Network with Discriminative Features"

_life, 2023, doi:10.3390/life13112153_

Round 1
Reviewer 1 Report
Comments and Suggestions for Authors
The authors proposed a DNN-based model for the identification of sumoylation sites using half-sphere exposure (HSE) measures for feature extraction and principal components analysis (PCA). Using PCA can lead to some loss of information if the right number of principal components that explain enough variation in the dataset is not selected. The authors did not explain or support their choice of 80 in the manuscript. Also, when PCA is applied on a feature set, the resultant loadings for features with high variance will be large. Hence, principal components will be biased towards features with high variance, leading to potentially false results. The authors must address this issue.
Hyperparameters play a significant role in model development. Given the number of samples used in the current study, the authors must provide details of other hyperparameters. From Table 2, it seems that only tuned hyperparameters are learning rate and activation functions, which are not sufficient for model development. Details of other hyperparameters should also be provided.
For authentication and reproducibility, the authors must provide the source code on GitHub.
The comparison with recent literature using their independent dataset to show the generalization of the model is missing, e.g., ResSUMO 2022 (PMID: 36078053, PMCID: PMC9454673), SUMO-Forest (PMID: 32160959), iSUMOK-PseAAC (PMID 34430072)
To improve the readability of the manuscript authors must address the typos e.g. on page 4 second paragraph appropriate space in words is missing similarly on page 6 etc.
Author Response
Thank you for your valuable time and effort for thorougly reviewing our manuscript.

Reviewer 2 Report
Comments and Suggestions for Authors
The identified problem statement in the manuscript "Enhancing Sumoylation Site Prediction: A Deep Neural Network with Discriminative Features" is really good. I enjoyed while reading the manuscript but there are few points that needs to be seriously addressed before I recommend this manuscript to be published,
@ Are there any potential biases or limitations in the automated detection approach that should be addressed?
@ Are there any unexpected findings or patterns in the data that should be explored further?
@ The implications of the research for the field has to be explicitly discussed at the end of introduction.
@ The literature review looks to weak. The authors should have discussed in depth about so far research done using deep learning for different sites prediction. I will like to recommend authors to make a detailed paragraph in introduction section and discuss about following manuscripts recently being published using ML techniques for sites prediction. (Talking about different sites and different models will attract the readers attention)
- https://doi.org/10.1093/bioinformatics/btad474
- https://doi.org/10.1016/j.ymthe.2023.05.016
- https://doi.org/10.1080/07391102.2021.2011785
- https://doi.org/10.1093/bioinformatics/btac434
- https://doi.org/10.1093/bib/bbac599
-DOI: 10.1109/TCBB.2022.3192572
- DOI: 10.1109/ACCESS.2021.3054361
@ In the conclusion section please highlight your future works.
@ Shed light on how you selected different parameters of the neural network model.
Comments on the Quality of English Language
Its fine
Author Response

(The authors gave the same response as above.)

Reviewer 3 Report
Comments and Suggestions for Authors
This manuscript introduces a sophisticated deep neural network model, termed Deep-Sumo, designed for the accurate prediction of sumoylation sites within proteins. Sumoylation is a pivotal post-translational modification implicated in the regulation of numerous essential biological processes. The paper's primary contributions include: 1) Utilization of Half-Sphere Exposure (HSE) for feature vector representation of protein sequences, 2) Employment of Principal Component Analysis (PCA) for feature selection, and 3) Construction of a deep neural network classifier for sumoylation site prediction. The methodology is robust, incorporating HSE for feature extraction, PCA for dimensionality reduction, and a deep neural network for classification. Empirical validations adhere to starondard ptocols, including training/testing splits and cross-validation. Comparative analyses reveal superior predictive accuracy, making this work a noteworthy addition to the domain of sumoylation site prediction.
Suggested Revisions:
-
Elaborate on the specifics of model training, including the number of epochs, batch size, optimization algorithms, and regularization techniques employed, to ensure complete reproducibility.
-
Conduct ablation studies to scrutinize the contributions of individual components, such as evaluating the model's performance in the absence of PCA-based feature selection.
-
Address the issue of class imbalance in the original dataset by considering more sophisticated balancing techniques beyond simple undersampling.
-
Justify the selection of evaluation metrics, providing a rationale for the appropriateness of metrics like accuracy and MCC for this specific research context.
-
Expand the discussion and conclusion sections to offer deeper insights into the limitations, challenges, and future directions of the study.
Minor editing of English language required.
Author Response

(The authors gave the same response as above.)

Round 2
Reviewer 1 Report
Comments and Suggestions for Authors
In order to demonstrate the proposed method's generalization and effectiveness, it is crucial for the authors to compare it with existing methods. The most common and fair way to do this is to save the model/weight of the proposed architecture and evaluate the method using independent datasets available in existing methods. This will help ensure a fair and accurate comparison between the proposed method and the existing ones.
Author Response
Please find the attached response and manuscript files.
Thank you
